# User experience with pregnancy tracker mobile apps: Findings from comment-based qualitative study

**Md. Alamgir Hossain**[1,2]*, **Janet Perkins**[3], **Md Tanvir Hasan**[2], **Sharon Low Bee Lian**[2], **Ribica Chowdhury**[2], **Md. Tanzirul Alam**[2], **Anisuddin Ahmed**[1,4], **Mohammad Sohel Shomik**[1], **Shams El Arifeen**[1], **Ahmed Ehsanur Rahman**[1,3], **Aniqa Tasnim Hossain**[1], **M. Shafiqur Rahman**[5]

1 Maternal and Child Health Division, International Centre for Diarrhoeal Disease Research, Bangladesh, Dhaka, Bangladesh, 2 BRAC James P. Grant School of Public Health, BRAC University, Dhaka, Bangladesh, 3 National Institute for Health and Care Research Global Health Research Unit on Respiratory Health, Usher Institute, University of Edinburgh, Edinburgh, United Kingdom, 4 Global Health and Migration Unit, Department of Women's and Children's Health, Uppsala University, Uppsala, Sweden, 5 Institute of Statistical Research and Training, University of Dhaka, Dhaka, Bangladesh

* alamgir.hossain@icddrb.org

**Data Availability Statement:** The data underlying the results presented in this study are available from websites of the pregnancy tracker apps. In this study we have use open for a third party and

## Abstract

### Background

Worldwide, millions of pregnant women use pregnancy-related apps to monitor their baby's growth and development. While most of the apps are user-friendly, not all of them are equally appealing. This study aimed to explore the user experience (UX) of pregnancy tracker mobile apps used by pregnant women.

### Methods

This study explored the dynamics between users' experiences and multifaceted dimensions of advanced features, high-quality materials and information, strict privacy policies, problem-solving abilities, and the usefulness of app features and contents. This study applied reviewers' comment-based qualitative study, accessing crowdsourced data gathered from different pregnancy tracker app websites. A thematic and content analysis approach was used.

### Results

This study found that when users are satisfied with using advanced content and features, it aligns with their perceived self-righteousness and rationality, and reflects their cultural values and expectations of using the apps. Conversely, when users encounter challenges such as erroneous baby size comparison and app updating issues, they perceive these as disadvantages of the apps utilised. Moreover, the study sheds light on the specific desires of pregnant women, highlighting their expectations for content that addresses their physical and mental well-being, as well as their unborn babies. The desire for free access reflects the cultural emphasis on cost-effectiveness, while the willingness to invest financially in enhanced experiences demonstrates the recognition of the value and potential benefits of

data available in public domain below mentioned pregnancy tracker apps websites. The URL of the apps are share in the manuscript.

**Funding:** The author(s) received no specific funding for this work.

**Competing interests:** The authors have declared that no competing interests exist.

improved content and features. This study also provides valuable insights into the complex relationship between users' experiences, cultural values, advanced features, high-quality materials and information, privacy policies, problem-solving abilities, and relevant content in creating positive app experiences that align with users' cultural expectations and needs.

## Conclusion

This study provides essential insights into the user experience and underscores the importance of a user-centric design approach for developers. By capturing the current landscape of these digital tools, can provide valuable feedback for the enhancement of existing applications and guide the development of future iterations, ensuring the diverse preferences and expectations of pregnant women worldwide.

## Introduction

Annually, unwanted pregnancies totaled 121 million in 2015–19, which means a global rate of 64 unplanned pregnancies per 1000 women aged 15–49 [1]. This number includes both intended and unintended pregnancies, regardless of the outcome, such as live birth, abortion, or miscarriage. Unfortunately, many pregnancies result in negative outcomes. As reported by WHO, a total of 2,87,000 maternal deaths were reported globally in 2020 [2, 3]. Unavailability or inaccessibility of antenatal care, inadequate maternal nutrition, low educational status, and lack of health literacy among women significantly influence women's health-seeking behaviour during pregnancy [4–6]. Globally, pregnant women use digital health tools such as mobile apps to obtain pregnancy-related information and expect quality content from health experts in mobile apps [7, 8]. Pregnancy-related apps are useful for expectant mothers to obtain information about nutrition, physical activity, and bodily changes [9–11]. Different pregnancy tracking apps are found useful, and offer different features, such as monitoring weight gain and foetal development throughout the pregnancy period, keeping track of the length of contraction, and providing reminders for medical appointments and medication [12].

With the advancement of technology, digitised solutions have become popular to replace in-person medical visits and seek healthcare services. For smartphones, there are more than 259,000 health-related apps accessible in app stores (such as Google Play and Apple's iOS "app store"). As of early 2015, there were about 100,000 m-health apps available [13, 14]. There are many benefits to using mobile health apps. They can access evidence-based health information, monitor health parameters, self-manage chronic conditions, and improve medication adherence [15]. These apps can give end users worldwide, 24/7, affordable access to high-quality, evidence-based health information [16]. Studies found that the benefits of mobile apps were greater than the challenges in terms of personalised care like self-management, personalised exercise and monitoring, and helping friends and family avoid harm in dealing with stroke of elderly people [17]. Mobile apps have changed how we understand and experience digital interactions [9]. There was a 25% increase in the number of downloads of mobile health apps during the COVID-19 pandemic compared to the same time the year before [18]. Even before the pandemic, pregnancy apps were more commonly used than other health-related apps [19].

Based on the available data, many pregnant women use digital media sources, including pregnancy tracker apps, to access information and support during their pregnancy journey.

We also have insights into the popularity of pregnancy apps and their potential to promote healthy behaviours and self-monitoring [20]. Additionally, studies have highlighted the interest and confidence of women in using apps for pregnancy-related purposes [21]. However, users focus on design and functionality over evidence quality and author credentials when assessing app credibility, heavily influenced by popular opinion and peer reviews [22]. Although many apps are being developed to support pregnant mothers in tracking pregnancy outcomes, most of them raise concerns about standardisation, accessibility, reliability, and quality [10].

Hundreds of apps are available but very few of them were developed following rigorous scientific procedures [23]. Therefore, it is necessary to ensure the quality of these apps to provide pregnant women with a better experience of their pregnancy. However, few studies have explored the quality of the existing pregnancy apps through users' review opinions.

Comment-based qualitative study is becoming more widely acknowledged as a useful and cutting-edge research approach that offers several special benefits to the study of human behaviour, culture, and social interactions in the digital sphere [24–27]. It enables researchers to explore behaviours and cultural practices in the digital world and users' experiences in virtual communities and online platforms [28–30]. Comment-based qualitative studies can provide further knowledge and understanding in several areas. First, it can offer real-time data on how women use pregnancy tracker apps, their behaviours, and interactions within these digital platforms. This can help uncover the specific features and functionalities that users find valuable and the challenges they may encounter. Second, a comment-based qualitative study can shed light on the quality of content and self-monitoring tools within pregnancy apps, as well as the inclusion of behaviour change techniques. This information can inform the development of more effective and user-centered apps. Third, comment-based qualitative studies can explore women's perceptions, recommendations, and preferences for pregnancy apps, including desired features such as personalised support and integrated care. Finally, the scientific community and app developers can understand why some pregnancy tracking apps are more appealing compared to other available apps. Therefore, we aimed to conduct a comment-based qualitative study to explore the user experience (UX) of the pregnancy apps used by pregnant women. By understanding users' interactions within online communities and considering the broader social dynamics, researchers can uncover the social, cultural, and emotional aspects that shape the user experience on pregnancy tracker apps.

## Materials and methods

### Study design

This research employed a comment-based qualitative study. In this study, we used thematic analysis and content analysis approaches that allowed the study to explore the user experience with pregnancy tracker mobile apps comprehensively and systematically, drawing on real-world data and identifying important themes and patterns. The use of comment-based qualitative data enabled the study to capture a wide range of experiences and perspectives in a digital context. We extracted user review and comment-based data from different app websites and analysed them to understand both the positive and negative experiences of the users, their expectations of the apps, and their reasons for recommending them to others.

### Settings

This study was conducted on the Google platform. The data were obtained from different pregnancy tracker app websites.

**Table 1. Types of news articles, blogs, and websites.**

| News articles | **forbes.com, abcnews.go.com, washingtonpost.com** |
|---|---|
| Journals | ncbi.nlm.nih.gov |
| Blogs | blog.ohiohealth.com |
| Websites | glamour.com, motherandbaby.com, whatexpect.com, cosmopolitan.com, goodhousekeeping.com, noodleandboo.com, today.com, babylist.com, babyheart.com, top10.com, makeuseof.com, androidauthority.com, bestproduct.com, momjunction.com, apiumhub.com, emmasdiary.com, netmums.com, my-best.com, and parents.com. |

## Data collection

Data used in this study were extracted from secondary sources which are available in the public domain. These include pregnancy apps used by the general user on the Google platform. The pregnancy tracker apps were selected through a systematic search on the Google platform. We used the following search items: "Pregnancy Tracker Mobile Apps", "Pregnancy Apps" and "Pregnancy Tracker Apps". We searched for the names of pregnancy tracker apps in journals, news articles, blogs, and websites (Table 1). We used purposive sampling procedures to select the review comments provided by the pregnancy tracker mobile app users.

## Inclusion and exclusion criteria

Inclusion and exclusion of the apps screening were conducted by two team members (MAH and MSR). This study included only English-language pregnancy tracker mobile apps that mentioned "pregnancy" in the app's name. This study considers global apps rather than prioritise regional or national geographical locations. The exclusion criteria for the apps were not mentioned pregnancy apps, ratings below 4.2, no comments on the app's website, no content rating and content rating below 3+, and many ads mentioned in app's description (Fig 1). Review and comment-based data were extracted from the different websites of the apps between July 2023 to October 2023.

## Data filtering

This study collected the review comment-based data following the content analysis approach and later coded it based on the theme. Two-member team (MAH and MSR) reviewed each user-review comment by visiting the websites of each selected app (S1 Fig). A total of 426 review comments (of pregnant women and their partners) were extracted from the websites of the ten apps through the Google Play Store (Android) and App Store (iOS). The comments were selected based on the relevance of the research objectives. Team members filtered out irrelevant comments according to the research objectives and short comments that were unable to express the meaning of the objectives. For analysis, a total of 120 comments were considered on positive experiences, 96 comments on negative experiences, 91 comments on expectations, and 119 comments on app recommendations.

## Data analysis

To analyse the review comments qualitatively, thematic analysis were used. The steps of the data analysis procedure are depicted in Fig 2. At the initial stage, two team members (MAH and MSR) read the reviewers' comments line by line. Moreover, a coding matrix were developed to conduct a qualitative textual analysis. Additionally, data were coded based on the objectives of the study, such as i) users' positive experiences related to using pregnancy apps,

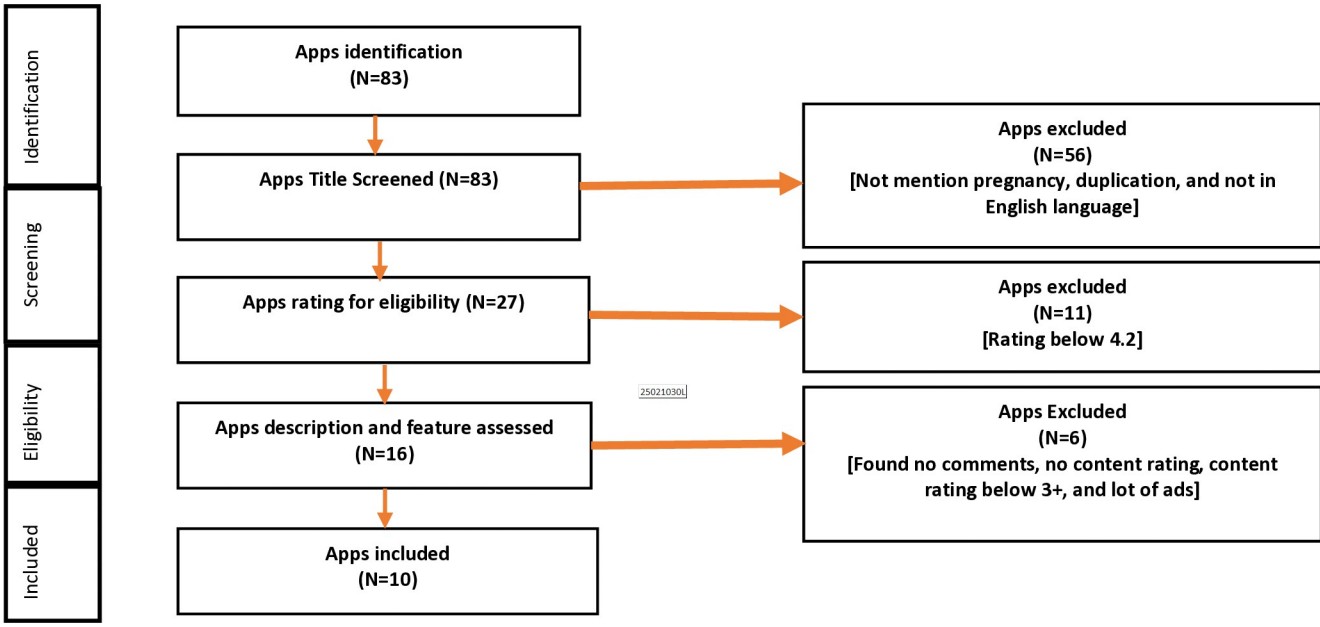

**Fig 1. Flow chart of the app's inclusion and exclusion.**

ii) users' negative experiences related to using pregnancy tracker apps, iii) expectations related to pregnancy tracker apps, and iv) perceived reasons for recommending pregnancy apps to others. Furthermore, researchers independently analysed the coded data to understand the data based on the study objectives until a consensus was reached. Finally, results are interpreted thematically. Thematic analysis is a qualitative research method that identifies recurring patterns or themes in data through thorough examination and interpretation to extract meaning and understand different subjects and perspectives [31–33]. During the data analysis, research team members arranged online meetings weekly to discuss the findings and understand the correct meaning of the data.

In this study, each team member brings their unique background, experiences, and perspectives, which can influence how data are perceived and analysed. This collective awareness allows us to critically engage with the data, ensuring that our interpretations reflect the users' experiences rather than solely shaped by our viewpoints. The critical examination of reviewer

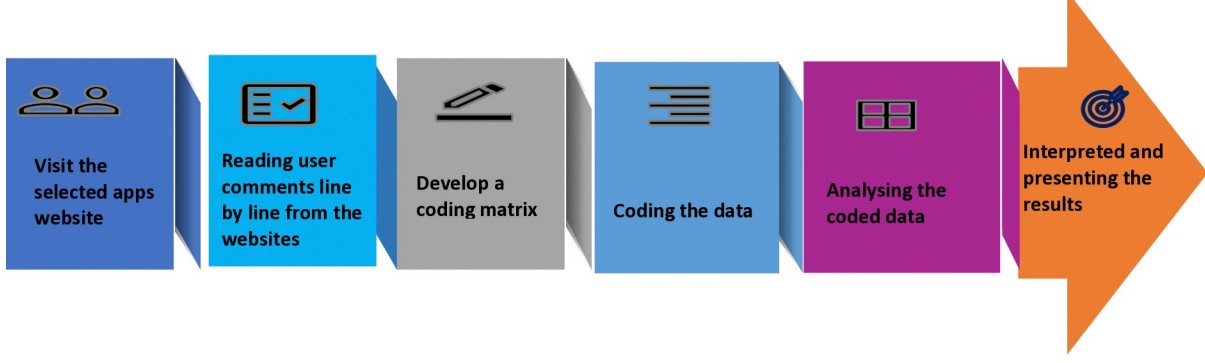

**Fig 2. Steps of comments-based qualitative data analysis.**

comments and user perspectives is essential in qualitative research, particularly in the context of the digital world, particularly in the pregnancy tracker mobile apps (S1 Table). By engaging with the comments, the team reviewed each comment critically, to uncover biases, assumptions, and underlying themes that may not be immediately apparent. This analysis allows us to recognise the influence of our positionality such as our experiences, beliefs, and social identities on the research process. During the data analysis, researchers were careful to interpret the extreme comment-based opinions.

## Ethical statement

This study was conducted for partial fulfilment of the executive certificate course on 'Population Health Informatics'. The exam board members oversee the ethical and research-related issues and approve the proposal and thesis from the BRAC James P Grant School of Public Health, BRAC University. The research conducted does not involve human participation and relies on third-party data sourced from public websites. Since the data is secondary, ethical approval and informed consent are not required. The study ensures the anonymity of users and the app's names, thereby respecting privacy during data analysis. Moreover, this study ensures anonymity and confidentiality. For that reason, permission for data use is not necessary.

## Results

This study found a total of eighty-three mobile apps related to pregnancy. A total of ten apps relevant to the pregnancy of mothers and compatible with Android and iOS mobile phones were selected based on the inclusion and exclusion criteria. The names of the ten selected apps are Pregnancy+ (Android), Pregnancy+ (iOS), Pregnancy & Baby Tracker–WTE (iOS), Pregnancy Tracker–BabyCenter (iOS), OVIA Pregnancy & Baby Tracker (Android), Ovia Pregnancy & Baby Tracker (iOS), Pregnancy tracker- Preglife (iOS), HiMommy Pregnancy Tracker App (Android), HiMommy Pregnancy Tracker App (iOS), and Hello Belly: Pregnancy Tracker (iOS). For this study, ten apps were considered, among which three were Android apps and the remaining seven were from the iOS platform. The Table 2, consists of the name of the apps, compatibility, name of developers, privacy issues included or not, description of the apps, features and contents, years of starting services, total rating, content rating, number of reviewers, and number of downloads (Table 2).

### i) Perceived positive experiences of using different apps

Pregnant women who used pregnancy tracker apps had a useful experience with the features and content, they appreciated the apps by providing positive feedback such as reviewer comments and emoji or emoticons (pictogram or smiley imbedded in text and used in electronic messages). The features of the apps included showing the baby's growth, appearance, and growth development week by week, using images and size comparisons with an object, tracking birth plans, weight measurements, monthly baby photographs for progression, and a baby name option. Moreover, users feel amazed by obtaining 3D images, descriptions with illustrations and visualisation, physician appointments, and tracking for medical needs; thus, they provide higher ratings. A pregnant woman commented,

> "*This is a fantastic app that takes you through your pregnancy week by week. Explaining every vital aspect, such as how your baby is growing and how you should be feeling based on your pregnancy stage.*" (A user of pregnancy tracker apps on an Android phone)

**Table 2. Overview of the pregnancy tracker app.**

| SL | Name of apps | Compatibility | Developers | Privacy | Description of apps | Features and contents | Years of starting | Total rating | Content rating | Number of reviews | Number of downloads |
|---|---|---|---|---|---|---|---|---|---|---|---|
| 1 | Pregnancy+ | Android | Philips Digital UK Limited | Privacy policy included | Baby Development, Pregnancy Guides & Information, Pregnancy Tools, Organize & Plan, 3D Models, pregnancy articles, and guide and share journey with family and friends | Expert advice, daily articles, healthcare tips, interactive 3D models, baby's development | 2013 | Android: 4.4 out of 5 | 3+ | Android: 2.42 M | Android: 10 M+ |
| 2 | Pregnancy+ | iOS | Philips Digital UK Limited | Privacy policy included | Baby Development, Pregnancy Guides & Information, Pregnancy Tools, Organize & Plan, 3D Models, pregnancy articles, and guide and share journey with family and friends | Expert advice, daily articles, healthcare tips, interactive 3D models, baby's development | a | iOS: 4.8 out of 5 | 12+ | iOS: 57.5K | iOS: a |
| 3 | Pregnancy & Baby Tracker–WTE | iOS | Everyday Health, Inc. | Privacy policy included | Date calculator, medically accurate articles, daily tips, pregnancy weight, baby's feeding, postpartum recovery | Baby size comparisons, Due date calculator, weekly pregnancy tracker, daily tips, Expert-reviewed articles, Community groups | 2010 | iOS:4.9 out of 5 | 12+ | iOS:320.5k | iOS: a |
| 4 | Pregnancy Tracker–BabyCenter | iOS | Everyday Health, Inc. | Privacy policy included | BabyCenter's free app, parenting updates, Baby Growth Tracker, and sleep and feeding guides. | Pregnancy, parenthood, starting a family, baby centre community, pregnancy tools | 2011 | iOS:4.9 out of 5 | 12+ | iOS: 209.7K | iOS: a |
| 5 | Ovia Pregnancy & baby tracker | Android | Ovia health | Privacy policy included | Track baby's growth, due date calculator, baby growth calendar, symptoms, mood, sleep, activity, weight, blood pressure, nutrition, appointments, pregnancy milestones | Baby's anatomical development, 3D image, foetal movement, and nutrition serving | 2014 | Android: 4.7 0ut of 5 | 3+ | Android: 143k | Android: 1 M+ |
| 6 | Ovia Pregnancy & baby tracker | iOS | Ovuline, Inc | Privacy policy included | Daily and weekly updates, size comparison with a fun theme, pregnancy calendar, blood pressure and weight data, nutritional need, sleep exercise, medication, and vitamin | Baby names, realistic illustrations, baby size comparison, hand and foot size, 2000 + articles, food safety, and medications | a | iOS: 4.9 out of 5 | 12+ | iOS: 89.8K | iOS: a |
| 7 | Pregnancy tracker-Preglife | iOS | Preglife AB | Privacy policy included | Baby development, articles with detailed information, different podcasts, contraction timers, vaccination guides, and free online childbirth classes. | Pregnancy Calendar, baby development tracker, baby growth, photos organizer, pregnancy journal, video | 2011 | iOS: 4.9 out of 5 | 12+ | iOS: 1K | iOS: a |

*(Continued)*

**Table 2.** (Continued)

| SL | Name of apps | Compatibility | Developers | Privacy | Description of apps | Features and contents | Years of starting | Total rating | Content rating | Number of reviews | Number of downloads |
|----|----|----|----|----|----|----|----|----|----|----|----|
| 8 | HiMommy Pregnancy Tracker App | Android | a | Privacy policy included | Developing each day, pleasant words, bodily changes, child growth, preparation for childbirth | Daily notification, what to eat and do not eat, pregnancy journal, kick counter, bump pictures, weight tracker | 2014 | Android: 4.5 out of 5 | 3+ | Android: 16.6k | Android: 500k+ |
| 9 | HiMommy Pregnancy Tracker App | iOS | Idea Accelerator | Privacy policy included | Developing each day, pleasant words, bodily changes, child growth, preparation for childbirth | Daily notification, what to eat and do not eat, pregnancy journal, kick counter, bump pictures, weight tracker | a | iOS:4.8 out of 5 | 4+ | iOS: 2.8K | iOS: a |
| 10 | Hello Belly: Pregnancy Tracker | iOS | HelloBaby Inc | Privacy policy included | 3D and AR visualization, 400 tips, yoga video class, symptoms, hospital bag, nutrition | Expert tips, 3D and AR visualizations week-by-week, Yoga video classes, | a | iOS:4.2 out of 5 | 4+ | iOS: 1.9K | iOS: a |

• a information not available

Pregnant mothers express their appreciation for the ease of app use and high-quality photos of the uterus. Pregnant women often enjoy visualising their baby's growth, and the app enhances this experience by comparing the baby's size to various fruits and animals. For example, it features playful comparisons like a water bear in the first month, a pygmy seahorse in the second month, and a cardinal tetra fish in the third month, making the journey more engaging and fun. In addition, if the app included important content such as educating both partners and parents about body positivity, pregnancy myths, and the importance of maternal nutrition, the users liked it the most. In addition, if the app includes many handy features, such as to-do lists with suggestions, appointment trackers, and kick counters, the users also feel happy to use them. A user commented,

> "*This is my second pregnancy app, and I like it!" The materials are useful, the app is not ready, and the baby tracking is fantastic! I look forward to the weekly scans, and the information is quite beneficial." (A user of pregnancy tracker apps on an Android phone)*

Pregnancy tracker apps offer self-explanatory learning materials, diagrams for weight development, nutritional data, timely symptom alerts, and engaging information, often accessible without premium membership sign-ups, or payments. On the other hand, some users perceived pregnancy-related apps to be most useful when they were completely free as they did not have to pay any subscription fee to use the apps. In contrast, some users are willing to pay to obtain more reliable pregnancy-related information. A user commented,

> "*Nothing in life is free, and if you cannot afford it (app payment), it is a harsh insight, since babies are even more expensive." (A user of pregnancy tracker apps on iOS phone)*

Family members also use the apps to obtain updates about the baby and the mother. By using the app, they can guess the conditions of the baby and the mother and encourage the mother to lead a healthy lifestyle. A user commented,

*"I love knowing about my little Grandchild's development and getting tips that help me, encourage my daughter and refresh my memory!" (A user of pregnancy tracker apps on an Android phone)*

Advanced features of some apps connect pregnant women with their partners, where both may exchange daily information and send messages back and forth, creating a nice sharing moment among them. A user commented,

*"It truly sends me information on what is happening, and when it happens. I have this app, and my husband has the app too. It is linked to us, so we receive updates at the same time, and it pulls us closer together." (A user of pregnancy tracker apps on an Android phone)*

The app users prefer to maintain privacy strictly during app use, but the free apps may not offer it. During the app use, the users provide very sensitive personalised data such as pregnancy, date of delivery, baby's condition, weight, blood pressure and so on, which need to be protected. Therefore, some apps ask users to pay a small amount after the fourteenth week of pregnancy. However, with user-friendly features and content, app users prefer a strict privacy policy for personalised data use. A user commented,

*"I am a first-time mother in the United States, and I discovered X (app name) recommended online due to their strict privacy standards. I enjoy how it has so much amazing content from real professionals." (A user of pregnancy tracker apps on an iOS phone)*

Pregnant women like to show how the baby grows in the womb to their older children, considering it an amazing way to help their social-emotional development. Moreover, pregnant mothers can understand the progress of the babies by seeing the photos, and videos and they can use a contraction timer to track the frequency and duration of contractions during labour, and go to the hospital. In contrast, some apps have a feature named 'birth club' where pregnant mothers can chat with their peers, and they can share their pregnancy experience. In that club, pregnant mothers share their personalised feelings, experiences and reactions to the pregnancy. Therefore, pregnant mothers find it helpful to use 'birth clubs' because they can ask questions and advice, and they also receive different answers from experienced mothers. Sometimes, pregnant mothers get a supportive reply from their peers which keeps them calm and assures them not to worry. Such kinds of supportive discussion help to reduce their depressive condition and anxious attitude when complications arise. A user commented,

*"It (chat) always helps relieve anxiety when I have questions and get answers from other mothers who are going through the same things or know how you're dealing with." (A user of pregnancy tracker apps on an iOS phone)*

### ii) Perceived negative experiences of the users of different pregnancy tracker apps

The users of different types of pregnancy tracker apps raise their concerns on the app websites to inform other users about features and content. One common concern of app users was that

some apps were not updated periodically. Moreover, users of different apps encountered update-related issues and revealed that they followed the usual practice of closing the app, again resuming it, and sometimes restarting the phone again and again. However, even after doing all these things, the app did not work properly. The users encounter unexpected and irrelevant issues while using the apps, which is the reason for bad reviews with the titles "Disgusting software", and "DO NOT DOWNLOAD". These issues demotivated the users from using the app.

Users initially like to see their pregnancy status in the apps by providing mentioned information like last menstruation date, blood pressure, weight of mother, weakness, vomiting, etc. Based on the information provided for the apps, it gives positive or negative results. However, when they found false-negative results based on confirmatory pregnancy tests, that impacted their mental stress. A pregnant mother commented,

> "*It broke my heart! when I found I was not pregnant after the confirmatory test.".*" (A user of pregnancy tracker apps on an iOS phone)

Some users encounter visualisation issues such as a small font size that is not readable and visually disturbing background colours that affect the eyes. Moreover, some users encountered technological difficulties in obtaining 3D photos of the foetus. In some cases, the diary entry section and notes were completely lost after the apps were updated. Some apps do not show the actual weight data entered for a specific date. A user commented,

> "*The actual size of the baby in the 3D image, which does look very cool, is not accurate, and it is too large*" (A user of pregnancy tracker apps on an Android phone)

Some users revealed that the apps sometimes provide repeated pregnancy-related articles and materials rather than new and different types of important articles, such as morning sickness and pain. Repeating articles from the app's developer makes users unpleased. Moreover, pregnant women show dissatisfaction with the apps when they find very basic information on the apps. Similarly, the pregnant mother showed dissatisfaction when they did not find relevant information about the child's growth and pregnancy-related complications. In contrast, sometimes the information provided by the apps was inaccurate, rather than contradictory to the information provided by the physician, such as the due-date calculator for pregnant mothers. A user commented,

> "*Because the date calculator is quite inconvenient, it contradicts what your (users) doctor will calculate if you enter the right start of your LMP, plus the weeks are off.*" (A user of pregnancy tracker apps on an iOS phone)

A user revealed that the home page of an app is disorganised. Even after upgrading to premium, she failed to obtain the premium features. This app did not give her a pleasant experience in the second pregnancy as it did in the first pregnancy. A user commented,

> "*I used the app for my first pregnancy and hope it will work now as well. It keeps reminding me to change my due date. If I touch the back button accidentally, it indicates that my due date has passed. When I do not open it at least once a day, it reminds me to set a new due date!*" (A user of pregnancy tracker apps on an Android phone)

Users of pregnancy tracker apps mostly enjoy community message boards; however, some users have revealed that some posts in birth club make them nervous. In the birth club, users

ask for advice from their peers and discuss how to reduce their complexity during pregnancy. However, sometimes they receive worrying responses from their peers. Moreover, users believed that every pregnancy was different and tried to avoid opening "scary" posts. A user commented,

> "*I try to avoid opening "scary" posts that can make me feel nervous or freaked out. Even though I can see why someone with worries would post to a message board, I simply over-think!*" (*A user of pregnancy tracker apps on iOS phone*)

Users identified issues related to body-shaming of women during pregnancy because they were afraid that they would gain weight. Moreover, some phrases in the apps raised concerns about gender and were considered disrespectful to pregnant mothers, such as "people with wombs", "people with ovaries", and "chestfeeding". However, no "inclusive language" exists for men and women in pregnancy and birth, as only women may have experience. On the other hand, users gave three stars because of ineffective information, such as reminding them that 'sex is safe during pregnancy'. This kind of reminder raises concern among pregnant mothers because they provide very personalised information for women. Users perceived that sexual intercourse during pregnancy was a matter of the couple's own choice when the app reminded them that they felt ashamed as they knew that it was safe during pregnancy. In addition, some exercise-related videos are shared with the users. Users expected to have the option of watching what they wanted to watch without giving them the option to play any video considered disrespectful. A user commented,

> "*Please respect our WOMANHOOD. We are mothers, that is all.*" (*A user of pregnancy tracker apps on an Android phone*)

Users experience that information in an app is patronising, uncited, and sometimes inaccurate. For example, the Chinese Moon method for predicting gender was 90% accurate, but in reality, it was only 50% accurate. Some apps constantly remind users to avoid sugary foods, including natural sugar, to achieve weight management. The apps provided reminders such as "stay away from your desires and limit the amount of food you're eating to avoid gaining 'too much weight." Some apps mention that extra food is not needed for pregnant mothers. This information makes the user concerned about how weight gain is bad during pregnancy. However, mothers prefer safe and healthy baby births. A user commented,

> "*We don't need to be constantly reminded that sugar makes us fat, and it's okay to gain weight in places other than your tummy. They can worry about their physical shape after a baby is born safe and healthy.*" (*A user of pregnancy tracker apps on an iOS phone*)

### iii) Perceived expectations of pregnancy tracker app users

Pregnancy app users want antenatal non-invasive testing to allow for the early discovery of gender chromosomes. Furthermore, users expect calendar functionality in such apps because they are important for checking for gestational age at future dates and scheduling things such as travel and appointments ahead of time. Moreover, the users expect the baby's information rather than advice on foods during pregnancy. One user commented,

> "*Please provide me with more information about what my baby is doing right now, rather than things that seem unattainable.*" (*A user of pregnancy tracker apps on an Android phone*)

The users want features that can show a visual representation of the twin babies, as according to the claims of the users, there is no single software that specialises in twin pregnancy. On the other hand, one user preferred a dark theme while using the apps and relevant articles placed on the home page. Users who did not conceive naturally expected to calculate the dates of becoming pregnant based on the IVF-embryo transfer. A user commented,

> "*It would be nice to be able to calculate how far along you are in your pregnancy based on the IVF-embryo transfer day as well. Not everyone can conceive naturally." (A user of pregnancy tracker apps on an Android phone)*

Users want an option that allows them to erase or swipe content that is read or viewed. Some pregnant women anticipate receiving articles on likely Down syndrome in early pregnancy so that they can learn more about the disease. Moreover, first-time mothers want to nest ideas, tips, and storage solutions for baby clothes, clothing hacks, websites, and other useful things. The users of apps like to see the overall weight, moods, symptoms, and womb picture, but they are not willing to change the feature rate they like to compare simultaneously. A user commented,

> "*I'd like to be able to view my overall weight, moods, symptoms, and womb pics (photos). I have to go in every day to see these things. I'd like to be able to compare my photos side by side with information. (A user of pregnancy tracker apps on an Android phone)*

### iv) Perceived reasons for recommending to others to use the pregnancy tracker apps

Users recommend the apps to others who have fewer or no advertising issues and who have descriptive information. Additionally, as the apps help monitor the growth and development of the foetus, users would like to refer to other pregnant women. When users discover valuable features and content, they recommend the apps to others. For example, users enjoy the community tab function since it allows them to share their concerns with other users by posting their questions, or they can follow the posts of others. A user commented,

> "*My favourite feature of the app is the community tab—if I have a question about something, I write a post or see previous postings that were about the same issue. I would recommend this app to anyone pregnant and family members to track their progress!" (A user of pregnancy tracker apps on an iOS phone)*

The users want extended options on the apps that would allow them to record all appointments, assist with packing, and even birth plans. Whenever users find an app useful, they feel comfortable recommending it to others. A user commented,

> "*I have been using it for a long time, and I find it informative. It has made our pregnancy journey quite easier. Highly recommended!" (A user of pregnancy tracker apps on an Android phone)*

Pregnant mothers who use pregnancy tracking apps perceive that it helps to establish a spiritual connection with the unborn child by monitoring growth and development. They feel they can see the child's growth in their womb and talk to their unborn child. Furthermore,

users appreciated the app's inclusion of small animated videos on real pregnant women and live vaginal births. A user commented,

> "*I adored it! They used to include brief animated films showing the baby in the belly and what the baby was doing, as well as videos of real pregnant women. At the 40-week milestone, they had a live actual vaginal birth, which I enjoyed seeing!*" (A user of pregnancy tracker apps on an iOS phone)

## Discussion

This study found, that users' positive experiences depend on the usefulness of the features and contents of the apps. Advanced features, quality information and materials, strict privacy, and the ability to solve problems make an app appealing to users. Moreover, users' negative experiences depend on encountering update-related issues, problems with functionality and visualisation, technological glitches, the sending of repeated articles, scary posts from peers, gendered issues, and misinformation. Additionally, users' expectations of apps are based on their advanced features and content, such as non-invasive testing, IVF-embryo transfer and visual representation of twin babies and scientifically sound articles. Furthermore, users recommend the app to others when they find fewer advertisements, birth plans and monitoring growth charts, which help expectant mothers establish a spiritual connection with their unborn baby. Finally, pregnancy apps mainly provide desirable features such as data storage, web functionality, personalized tools, and social media integration. This study provides important insights and explains a wide range of thoughts and feelings, including annoyance, concerns, expectations, and usefulness, of pregnancy tracker apps shared by a group of users. Interestingly, useful features and content of the apps impact the users' overall satisfaction and positive experiences. Sometimes it motivates others to use apps (Fig 3).

This study found that pregnancy tracker app users are satisfied with using the apps. Another study found the contrast finding revealed that the overall quality of pregnancy

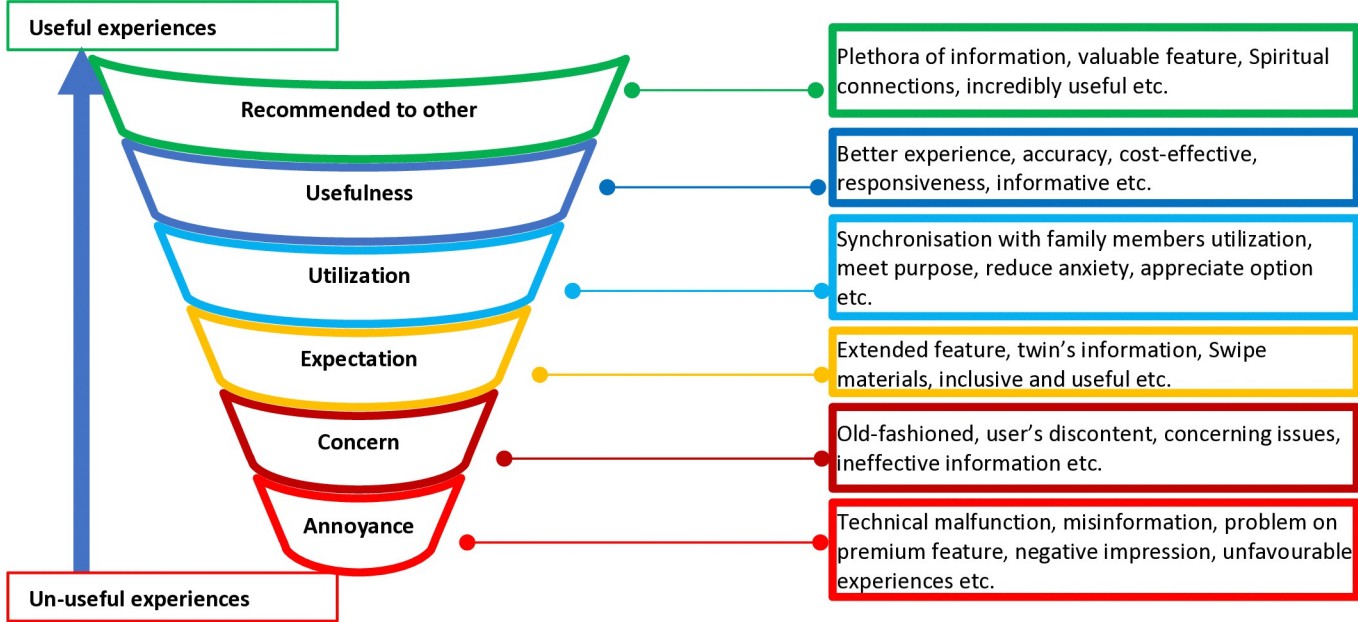

**Fig 3. Funnel of users' experiences using pregnancy tracker apps.**

tracking apps is moderate, with the highest scores in functionality and aesthetics based on the mobile app rating scale. Apps incorporate behaviour change techniques (BCTs), mainly focusing on providing information and instructions. However, most lacked effective BCTs for promoting physical activity, with common techniques including social comparison and self-monitoring prompts [34]. However, the unregulated health app industry, driven by commercial interests, hinders information reliability. Health professionals and users prefer culturally relevant and trusted pregnancy apps through the engagement of health professionals and institutions during app development [10]. There are over 350,000 digital health apps available in app stores, with approximately 250 added daily [35].

This study revealed that pregnancy tracker apps offer valuable features and content which is appreciated by users. Similarly, useful features for users include tracking a baby's growth and development, the use of pregnancy apps for Australian women to track foetal development and changes, and the use of all pregnancy-related information, such as daily health recommendations (21–23). Users emphasised the importance of the apps' capacity to provide comprehensive and reliable information. These resources address a variety of issues, including prenatal care, nutritional information, physical activity, and foetal development. Users appreciated the convenience of having access to a useful information in one centralized platform, which eliminated the need to search for information from multiple sources. This feature of pregnancy tracker applications not only saves time but also ensures that users have access to reliable and up-to-date information, increasing their confidence in making informed decisions.

This study found that good experiences encourage them to provide the highest rating and to recommend it to others. In contrast, users experience technical problems with apps that discourage them from using them. A study conducted in China revealed that pregnancy apps are useful and convenient for pregnant women because they are helpful for them to lead a healthy lifestyle during pregnancy. However, some apps deliver inaccurate and misleading information that raises concerns about their effectiveness. Pregnant women expect evidence-based, well-informed, tailored health information in apps [11]. Expectant mothers rely heavily on pregnancy tracker apps to access relevant and up-to-date information about their pregnancy journey. Moreover, users are ready to pay money if improved content and features are added to the apps. A review study (conducted in Australia) revealed that users were concerned about hidden expenses, such as in-app purchases and the requirement for internet connectivity to function properly, in 43% of the apps, but these concealed costs have the potential to delude customers [36]. The desire for free access to pregnancy tracker apps is not surprising, as users often expect digital literacy resources to be available without any monetary investment. This expectation is driven by the abundance of free apps available on the market, leading users to assume that pregnancy tracker apps should also be freely accessible. However, our study suggests that users are willing to reconsider this expectation if they perceive added value in terms of improved content and features. This study shows that users are not happy with some of the apps since these apps provide misleading information regarding a baby's sex based on the Chinese gender predictor chart, weight gain during pregnancy, and what to do and not do during pregnancy. Health practitioners and health organisations recommend providing authentic and trustworthy maternal health information through apps [37, 38]. In one study, women reported barriers to finding accurate information because they require multiple searches and sometimes received inaccurate information [10]. To address these issues, app developers should consider implementing rigorous content review processes and quality assurance measures. This can involve collaborating with healthcare professionals, conducting regular audits of the information provided, and remaining updated with the latest research and guidelines in the field of

prenatal care. Additionally, incorporating user feedback and reviews can help identify and rectify any inaccuracies or misleading information present in apps.

## Strengths and limitations of this study

This study has strengths and limitations. Comment-based qualitative study lies in its ability to gather detailed personal narratives and subjective experiences from users, allowing researchers to understand app usage's emotional and practical aspects. Moreover, researchers can identify gaps or unmet needs in existing pregnancy tracker apps by examining users' reviews and comments and informing developers on potential areas for improvement. Additionally, this study facilitates a deep understanding of users' lived experiences and perceptions to uncover nuances in user behaviour that may not be apparent through quantitative methods alone. Furthermore, this depth of information helps to reveal specific points, preferences, and suggestions for app improvement, and enhance user satisfaction. This study also has limitations. The comment-based data obtained from the app websites may not be representative of all users. This study may miss the perspective of those who do not engage in discussion on app websites. Moreover, those who leave review comments on app websites may express extreme opinions. Therefore, to get a realistic meaning of the data from the app's websites, study team members conducted repeated reviews of the comments.

## Recommendations

Several recommendations can be proposed based on the findings of this study. To enhance user satisfaction and engagement with pregnancy tracker apps, it is crucial to prioritise user-friendliness and minimise technical issues. Moreover, designing apps with intuitive interfaces and addressing technical glitches will contribute to a smoother user experience. Additionally, the app developers need to add professionally generated articles on different pregnancy-related topics. Furthermore, the need to ensure gender-sensitive content concerning womanhood and addressing privacy concerns are essential for creating inclusive and secure platforms. In contrast, limiting advertisements within apps will prevent user distraction and promote a more focused and enjoyable experience for individuals navigating the pregnancy journey. Additionally, more, the findings contribute to enhancing the overall user experience, guiding future research and development efforts in the field of maternal health technology. Finally, this research informs future research on usability testing and design principles, contributing to a more nuanced understanding of how app design impacts user behaviours and health outcomes.

## Conclusion

This study reveals critical insights that have significant implications for both developers and researchers in the field of digital health. The diverse nature of user experiences highlighted in this study emphasises the need for developers to adopt a user-centric approach in designing and updating pregnancy tracker apps. This study serves as a snapshot, capturing the current state of pregnancy tracker apps. The findings provide valuable feedback that can guide the improvement of existing apps and the development of future iterations, ensuring that these digital tools align with the varying preferences and expectations of pregnant women globally.

## Supporting information

**S1 Fig. Comment-based data in app websites.**
(TIF)

**S1 Table. Standards for reporting Qualitative Research (SRQR) guideline.**
(DOCX)

## Acknowledgments

This study was conducted for partial fulfilment of the executive certificate course on "Population Health Informatics. We thank BRAC James P Grant School of Public Health, BRAC University and icddr,b for their institutional support. We especially thank the core donors of icddr,b the governments of Bangladesh, and Canada for their support.

## Author Contributions

**Conceptualization:** Md. Alamgir Hossain, Janet Perkins, Md Tanvir Hasan, Sharon Low Bee Lian, Ahmed Ehsanur Rahman, Aniqa Tasnim Hossain, M. Shafiqur Rahman.

**Data curation:** Md. Alamgir Hossain, Janet Perkins, Md Tanvir Hasan, Sharon Low Bee Lian, Ribica Chowdhury, Md. Tanzirul Alam, Aniqa Tasnim Hossain, M. Shafiqur Rahman.

**Formal analysis:** Md. Alamgir Hossain, Md Tanvir Hasan, Ribica Chowdhury, Md. Tanzirul Alam, Anisuddin Ahmed, Ahmed Ehsanur Rahman, Aniqa Tasnim Hossain, M. Shafiqur Rahman.

**Investigation:** Md. Alamgir Hossain, Anisuddin Ahmed, M. Shafiqur Rahman.

**Methodology:** Md. Alamgir Hossain, Janet Perkins, Sharon Low Bee Lian, Ribica Chowdhury, Md. Tanzirul Alam, Anisuddin Ahmed, Ahmed Ehsanur Rahman, Aniqa Tasnim Hossain, M. Shafiqur Rahman.

**Validation:** M. Shafiqur Rahman.

**Writing – original draft:** Md. Alamgir Hossain, Janet Perkins, Md Tanvir Hasan, Shams El Arifeen, Ahmed Ehsanur Rahman, Aniqa Tasnim Hossain, M. Shafiqur Rahman.

**Writing – review & editing:** Md. Alamgir Hossain, Janet Perkins, Md Tanvir Hasan, Sharon Low Bee Lian, Ribica Chowdhury, Md. Tanzirul Alam, Anisuddin Ahmed, Mohammad Sohel Shomik, Shams El Arifeen, Ahmed Ehsanur Rahman, Aniqa Tasnim Hossain, M. Shafiqur Rahman.

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
