## [Decision Letter · Decision Letter 0]

30 Oct 2024

PONE-D-24-35703User Experience with Pregnancy Tracker Mobile Apps: Findings from Comment-based Qualitative StudyPLOS ONE

Dear Dr. Hossain,

Thank you for submitting your manuscript to PLOS ONE. After careful consideration, we feel that it has merit but does not fully meet PLOS ONE’s publication criteria as it currently stands. Therefore, we invite you to submit a revised version of the manuscript that addresses the points raised during the review process.

We look forward to receiving your revised manuscript.

Kind regards,

Ankit Anand, PhD

Academic Editor

PLOS ONE

Journal Requirements:

2. In your Methods section, please include additional information about your dataset and ensure that you have included a statement specifying whether the collection and analysis method complied with the terms and conditions for the source of the data.

“This study was conducted for partial fulfilment of the executive certificate course on population health informatics. The course program is funded by the Open Society University Network. We thank BRAC James P Grant School of Public Health, BRAC University and icddr,b for their institutional support. We especially thank the core donors of icddr,b the governments of Bangladesh, and Canada for their support.We note that you have provided additional information within the Acknowledgements Section that is not currently declared in your Funding Statement. Please note that funding information should not appear in the Acknowledgments section or other areas of your manuscript. We will only publish funding information present in the Funding Statement section of the online submission form.”

4. For studies involving third-party data, we encourage authors to share any data specific to their analyses that they can legally distribute. PLOS recognizes, however, that authors may be using third-party data they do not have the rights to share. When third-party data cannot be publicly shared, authors must provide all information necessary for interested researchers to apply to gain access to the data. (https://journals.plos.org/plosone/s/data-availability#loc-acceptable-data-access-restrictions)  

Additional Editor Comments:

The manuscript provided an interesting content analysis. However, there are multiple writing related issue raised by reviewer including adequate references.

The authors need to work on improving the writing and flow of the analysis as expressed in the reviews.

Reviewers' comments:

Reviewer's Responses to Questions

**Comments to the Author**

1. Is the manuscript technically sound, and do the data support the conclusions?

Reviewer #1: Yes

Reviewer #2: Partly

2. Has the statistical analysis been performed appropriately and rigorously? 

Reviewer #1: N/A

Reviewer #2: I Don't Know

3. Have the authors made all data underlying the findings in their manuscript fully available?

Reviewer #1: No

Reviewer #2: Yes

4. Is the manuscript presented in an intelligible fashion and written in standard English?

Reviewer #1: No

Reviewer #2: No

5. Review Comments to the Author

Reviewer #1: PONE-D-24-35703

User Experience with Pregnancy Tracker Mobile Apps: Findings from Comment-based Qualitative Study

Thank you for the opportunity to review this manuscript. The authors present findings from a comment-based qualitative study where they evaluated user experience with pregnancy tracker mobile apps. This manuscript deals with an important issue in e-Health/mobile app user perception, usage, and interaction. Overall, they find evidence to support positive user experiences, but also negative experiences of using the different apps.

I enjoyed reading the manuscript; however, there were some sections that were difficult to interpret and that I suggest should be re-written for better reader comprehension (see specific areas below). Similarly, there are some spelling and grammar mistakes that overall do not affect the understanding of the manuscript, but that should be addressed to meet a satisfactory standard for a journal publication.

I am proving here some comments that could potentially improve the understanding of the paper:

Abstract:

The first 3 lines in the results section should go in the methods section “the study explores the dynamics between ….” as it mentions the methodology. Line 44 in the results section seems to be incomplete: “When users are satisfied with using advanced content and features that align with their perceived self-righteousness and rationality, mirroring the cultural values and expectations”. I suggest that it is re-revised and re-written.

Main manuscript:

Background: lines 68-69. The reference provided is 10 years old and the relative number of pregnancies would be inaccurate. Please update accordingly.

Similarly, the statistics reported on lines 71-72 could be updated to represent more current data on maternal deaths.

Lines 72-75: the sentence: “Some of the factors responsible for adverse maternal health outcomes, though not limited to the unavailability or inaccessibility of antenatal care [3], inadequate maternal nutrition [4], low education status, and lack of health literacy among women, all of which affect their health-seeking behaviour during pregnancy “ is confusing. Please re-write it.

Line 77: do you mean women seek professional advice? I would not think that they use a pregnancy to seek a consultation with a health professional, as most of these free apps do not provide that service.

Line 85: reference is from 2013, please update it. Line 89: please provide context for this sentence: “The benefits of mobile apps were greater than the challenges”

Lines 105-109: please provide a reference(s) to support these statements.

Line 116: the sentence “comment-based qualitative can explore…” is missing the word ‘studies’. I suggest rephrasing to “comment-based qualitative studies can explore…”

Methods:

Line 136. Data are plural. Please correct the sentence to ‘data were obtained…”

Line 146: these findings are part of the results, and as such, should go into the results section of the manuscript. The inclusion and exclusion criteria should precede this.

Line 172: it should read ‘to analyse’

Line 183: it should read ‘how data are perceived and analysed”

Results:

Line 203: a short description of table 2 would be useful for the reader.

Lines 222-223: consider removing or rephrasing.

Line 251: but the free apps may not ‘offer’ it?

Line 284-288: I do not understand these sentences and how they relate to app issues.

Discussion:

Lines 411-414. How was this direct link established? These results should be introduced in the result section. No new information/findings should be presented in the discussion section.

Overall, the discussion should not introduce new information that has not been mentioned before. Please add these new findings to the results section.

strengths and limitations:

Line 476-476. Please review this sentence and use academic language "Therefore, we have to review it again and again to get a realistic meaning of the comments"

Reviewer #2: Thank you for allowing me to review the manuscript entitled 'User experience with pregnancy tracker mobile apps: Findings from a comment-based qualitative study. The research is an interesting topic and has merit. However, there are some major concerns with the manuscript as it currently presents. Examples of such are outlined below:

- Overall, the quality of writing is waivers in academic tone. For example, the consistent use of 'we' and the overuse of 'firstly', 'secondly', 'thirdly'... For a more academic tone, I would encourage avoidance of first-person language and, rather than listing points in so many of your sections, aim for a more fluent 'storyline' approach.

- I would also suggest a review of language flow and cohesion across the entire manuscript, as there were many instances of disjointed and confusing sentence structure. One example is in the Results section (p10, line 219): "Pregnant mothers are those who use the pregnancy tracker apps and express appreciation...". Pregnant mothers are those who are with child, regardless of their pregnancy tracker app use and appreciation. A second example (p11, line 239-240) "By using the app, they can guess the conditions of the baby and mother...". I would suggest reviewing this to say, "By using the app, they may gain insights into how the pregnancy is developing...", rather than imply that it is a guess. However, I do think the whole manuscript might benefit from proofreading revision.

- The abstract information doesn't seem to completely align with the abstract headings. For example, the first sentence under Abstract Results, should be in the Abstract Methods section. I also think the Abstract Conclusion could be more focused.

- In the Background section, there are non-formatted (line 68) and/or missing citations throughout (particularly the fourth paragraph [lines 106-123]). I would also encourage you to view pregnancy app reviews by Hayman et al. (2021, 2022) and also the pregnancy influencer study by Hayman et al. (2023), as they may be useful to consider within your background and/or discussion sections.

- The Methods section does not clearly articulate how many and who conducted the search and inclusion/exclusion screening, or data filtering - or if it was done independently. There were also shifts in language tense (past/present/future) that should be addressed throughout this section in particular. Also, a more digestible discussion of the thematic approach (with citations) would be useful to strengthen this section.

- Within the Results there are a few instances such as "...baby's size compared with the fruits and animals with fun stuff like..." (p10, line 221-222). The use of "fun stuff" (and similar) is not academic language. Please review the use of language to be highly academic tone.

- With the Results, under each thematic header, it might be worth structuring the details in a more logically flowing manner. In a few of the paragraphs, the focus seems to shift quite abruptly. A key example of this issue is on page 12 (line 275-293) where the focus starts on the issues caused by app updates, and then jumps to false-positive/negative experiences, then back to technical difficulties with imagery used. Perhaps using third-level headers might be helpful in further structuring the thematic focus points to flow more logically.

- On the point of false positives/negatives... Was the indication of pregnancy suggested by the app itself (i.e., was the app providing pregnancy results?) or was it that the user did a physical pregnancy test and simply shared her disappointment via the app community? If the latter, this point might be better placed in the community support results.

- In the Discussion (p16, line 411-412), you suggest there was a direct link between the usefulness of the apps and the user's overall satisfaction. However, from the analysis method and results presented, it is unclear how you were able to come this such a firm and distinct conclusion. Please revise to be less direct (e.g., 'there may be a link between...') or clarify how such a direct link emerged from qualitative analysis.

- Strengths and limitations... the final sentence needs review for appropriateness and academic language.

My broad review is reflective of the breadth of revision that I am encouraging before publication may be considered. I hope my above points can be useful in strengthening this manuscript.

6. PLOS authors have the option to publish the peer review history of their article (what does this mean?). If published, this will include your full peer review and any attached files.

Reviewer #1: No

Reviewer #2: No

---

## [Author Response · Author response to Decision Letter 0]

19 Nov 2024

Response to the reviewer's comments

We first express our gratitude to the editors and anonymous reviewers for providing your feedback and suggestions, which really enrich the manuscript. We try to address all your comments, and the following section provides point-point response to each comment.

---

## [Editor Report · Decision Letter 1]

8 Jan 2025

User Experience with Pregnancy Tracker Mobile Apps: Findings from Comment-based Qualitative Study

PONE-D-24-35703R1

Dear Dr. Hossain,

We’re pleased to inform you that your manuscript has been judged scientifically suitable for publication and will be formally accepted for publication once it meets all outstanding technical requirements.

Kind regards,

Ankit Anand, PhD

Academic Editor

PLOS ONE
---

## [Editor Report · Acceptance letter]

21 Jan 2025

PONE-D-24-35703R1 

PLOS ONE

Dear Dr. Hossain, 

I'm pleased to inform you that your manuscript has been deemed suitable for publication in PLOS ONE. Congratulations! Your manuscript is now being handed over to our production team.

Kind regards, 

on behalf of

Dr. Ankit Anand 

Academic Editor

PLOS ONE